# Synthesis of New Amino-Functionalized Porphyrins:Preliminary Study of Their Organophotocatalytic Activity

**DOI:** 10.3390/molecules28041997

**Published:** 2023-02-20

**Authors:** Pol Torres, Marian Guillén, Marc Escribà, Joaquim Crusats, Albert Moyano

**Affiliations:** 1Section of Organic Chemistry, Department of Inorganic and Organic Chemistry, Faculty of Chemistry, University of Barcelona, C. de Martí i Franquès 1-11, 08028 Barcelona, Spain; 2Institute of Cosmos Science, C. de Martí i Franquès 1-11, 08028 Barcelona, Spain

**Keywords:** asymmetric catalysis, Diels–Alder reaction, imidazolidin-4-ones, organocatalysis, photocatalysis, porphyrins, reductive amination

## Abstract

The design, synthesis, and initial study of amino-functionalized porphyrins as a new class of bifunctional catalysts for asymmetric organophotocatalysis is described. Two new types of amine–porphyrin hybrids derived from 5,10,15,20-tetraphenylporphyrin (TPPH_2_), in which a cyclic secondary amine moiety is covalently linked either to a β-pyrrolic position (Type A) or to the *p*-position of one of the *meso* phenyl groups (Type B), were prepared by condensation, reductive amination, or amidation reactions from the suitable porphyrins (either formyl or methanamine derivatives) with readily available chiral amines. A preliminary study of the possible use of Type A amine–porphyrin hybrids as asymmetric, bifunctional organophotocatalysts was performed using the chiral, imidazolidinone-catalyzed Diels–Alder cycloaddition between cyclopentadiene 28 and *trans*-cinnamaldehyde 29 as a benchmark reaction. The yield and the stereochemical outcome of this process, obtained under purely organocatalytic conditions, under dual organophocatalysis, and under bifunctional organophotocatalysis, were compared.

## 1. Introduction

The demand for the efficient production of enantiomerically pure chiral compounds has experienced an impressive growth in the past fifty years or so, driven particularly (but not exclusively) by the pharmaceutical industry. With its ability to afford structurally diverse sets of chiral molecules with high atom economies, asymmetric catalysis has played a central role in this challenge [1,2]. As a result of efforts in this area in the last two decades, asymmetric organocatalysis has become a well-established synthetic tool, together with enantioselective transition-metal catalysis and enzyme catalysis [3,4]. In approximately the same period, we have witnessed the development of visible-light-driven photocatalysis as a powerful synthetic method for performing radical transformations in a sustainable way, triggered either by triplet–triplet energy transfer (photosensitization) [5] or by single-electron transfer (photoredox catalysis) [6,7] between a photoexcited catalyst and the substrate. However, the interplay between photocatalysis and asymmetric catalysis is challenging due to the difficulty of controlling the stereochemistry of reaction steps that involve highly energetic radical intermediates. In the wake of the pioneering work of MacMillan, in a dual-catalysis approach involving the cooperative use of both a chiral organocatalyst and a photoredox catalyst [8,9], several successful, asymmetric photocatalytic transformations have been accomplished in the past years [10,11,12,13].

On the other hand, in an attempt to emulate the efficiency and the selectivity of enzymes, there is a growing interest in the development of bifunctional catalysis, in which a single molecule combines different catalytic activation modes [14,15]. Although it is conceptually more appealing, this approach also presents significant challenges. In the area of asymmetric organophotocatalysis, the pioneering work of Bach’s group [16] paved the way to further developments by Melchiorre [17] and Alemán [18].

While the best-known molecular photocatalysts are inorganic or organometallic coordination complexes of Ir and Ru, the continued interest in visible light photoredox catalysis has led to the increasing use of organic dyes as cheaper, less toxic, and more environmentally friendly photocatalysts [7,19]. Among these, porphyrins and their metalated derivatives have been extensively used as photosensitizers for singlet oxygen generation via energy transfer; however, they have been studied far less as photoredox catalysts [20]. In particular, the application of porphyrins as photoredox catalysts for carbon–carbon bond formation has only been recently explored by the Gryko group, both for the α-alkylation of aldehydes with diazo acetates [21] and for the photoarylation of π-excedent heterocycles with aryldiazonium salts [22]. More recently, de Oliveira reported on the utilization of fluorinated porphyrins as photoredox catalysts in the arylation of enol acetates [23]. In this paper, we wish to disclose in the design, synthesis, and initial study of amino-functionalized porphyrins as a new class of bifunctional catalysts for asymmetric organophotocatalysis.

## 2. Results and Discussion

### 2.1. Synthesis of Novel Amino-Functionalized Porphyrins

In the context of our research program on novel catalytic applications of porphyrins, [24,25,26] we have disclosed that the organocatalytic activity of both achiral [27] and of chiral [28] cyclic secondary amines linked to a 4-sulfonatophenylporphyrin scaffold can be efficiently regulated by the pH of the medium, which controls the aggregation state of the amphiphilic porphyrin moiety (Figure 1).

These amine–porphyrin hybrids were prepared through a mixed-porphyrin synthesis [29] from pyrrole, benzaldehyde, and a suitable *N*-Boc-protected amino aldehyde, in which, after oxidation with *p*-chloranil, a careful chromatographic purification is necessary to obtain the desired mono-functionalized porphyrins from the statistical porphyrin mixture in low yields (4–15%). In order to overcome this drawback, we decided to investigate the synthesis of two new types of amine–porphyrin hybrids derived from 5,10,15,20-tetraphenylporphyrin (TPPH_2_), in which the cyclic secondary amine moiety is linked either to a β-pyrrolic position (Type A) or to the *p*-position of one of the *meso* phenyl groups (Type B). As depicted in Figure 2, these two types of amine-functionalized porphyrins could be accessed from the known 2-formyl-5,10,15,20-tetraphenylporphyrin **1** [30] and from 5-(4-formylphenyl)-10,15,20-triphenylporphyrin **2**, [31] through condensation or reductive amination reactions with the suitable amines, taking advantage of the synthetic versatility of the formyl group.

2-Formyl-tetraphenylporphyrin 1 and its metal complexes have been used in several instances as convenient starting materials for the synthesis of *meso*-tetraphenylporphyrins directly linked with an heterocyclic moiety at the β-position, either through 1,3-dipolar cycloadditions [32,33,34,35] or by lanthanum(III) triflate-catalyzed condensation reactions [36,37,38,39]. It is worth noting, however, that 1 had not been employed for the preparation of chiral imidazolidinone–porphyrin hybrids. The first step in the path towards amine–porphyrin hybrids of Type A was therefore the synthesis of 1, involving the metalation and subsequent formylation of TPPH_2_. The methodology used in this step was described by Richeter et al. in 2002 [40]. For the metalation of the porphyrin core, copper(II) acetate was added to a refluxing TPPH_2_ solution in dichloromethane:methanol. After filtration, a purple solid was obtained with a 97% yield. Subsequently, the Vilsmeier–Haack formylation of the metalated porphyrin (TPPCu) was carried out, furnishing the copper(II) 2-formylporphyrinate complex **3** in a quantitative yield after hydrolysis with aqueous sodium acetate. In order to obtain the non-metalated porphyrin-2-carbaldehyde **1**, the crude Vilsmeier–Haack formylation mixture was treated with concentrated sulfuric acid before hydrolysis, according to the procedure of Bonfantini et al. [30] (Figure 1).

With compounds **1** and **3** in our hands, we set out to prepare the corresponding MacMillan imidazolidinones through the condensation of the formyl group with the *N*-methyl amides of the desired α-amino acids, according to the protocols described by Samulis and Tomkinson in 2011, which make use of ytterbium(III) triflate (YbTf_3_) as a Lewis acid catalyst [41]. To that end, we obtained the *N*-methyl amides **4a**, **4b**, and **4c**, derived from L-alanine, L-phenylalanine, and L-*tert*-leucine, respectively, by a two step-route that implies the treatment of the methyl ester hydrochloride (either commercially available or easily obtained by the esterification of the amino acid in a methanol solution in the presence of thionyl chloride) [42] with an excess of methylamine, followed by neutralization with NaOH in ethanol or with aqueous ammonia [41,43]. However, all our attempts to obtain imidazolidinones derived from **1** by direct condensation with *N*-methyl amides were unsuccessful; even when a solution of **1** and of **4a** (1.1 equivalents) in chloroform was heated at reflux in the presence of YbTf_3_ for 72 h, a mixture of starting aldehyde and the imine was obtained with no trace of the desired imidazolidinone **6a**. It is worth noting that under these conditions, Alemán and co-workers were able to achieve the formation of several thioxanthone-derived imidazolidinones from the corresponding aldehydes [18].

We reasoned that this failure could be due to the complexation of Yb(III) by the porphyrin core. In order to test this hypothesis, we submitted the Cu(II) complex **3** to the same reaction conditions. We were pleased to find that after 24 h, a 26% yield of the metalated imidazolidinone **5a** was obtained upon chromatographic purification on silica gel. A dichloromethane solution of **5a** was stirred for 4 min with concentrated H_2_SO_4_ (1 mL/mmol) at room temperature and poured over aqueous, concentrated NaOH at 0 °C. After extraction with chloroform, chromatographic purification afforded (with a 48% yield) the desired imidazolidinone **6a** as a single diastereomer, according to ^1^H NMR (Figure 2; see Appendix A in the SM). On the precedent of the previously observed stereochemical preferences of 2-substituted imidazolidin-4-ones, [18,41] we tentatively assigned a *trans* relative stereochemistry to **6a**.

On the other hand, when **3** was reacted with the methyl amide **4b**, derived from L-phenylalanine, and the crude product obtained (91% yield) was submitted to chromatographic purification, the Cu(II)-imidazolidinone **5b** was obtained with a 50% yield (as an unknown mixture of diastereomers, two spots by TLC). The demetalation of **5b** with concentrated sulfuric acid, followed by neutralization and filtration over a silica gel pad with dichloromethane:methanol 98:2, afforded the demetalated imidazolidinone as a mixture of the diastereomers **6b**/**6b’** with a 78% global yield. Careful chromatographic purification allowed us to isolate the pure diastereomers **6b** and **6b’** with an 18% and 26% yield, respectively (Figure 3). As is explained in more detail in the Appendix A, on the basis of their ^1^H-NMR spectra, a relative *cis* stereochemistry was assigned to the minor isomer **6b** and a *trans* stereochemistry was assigned to the major isomer **6b’** (See Appendix A in the Supplementary Information).

Finally, the condensation reaction between the Cu(II) 2-formylporphyrinate **3** and the *tert*-L-leucine-derived amide **4c** (1.3 equiv) was again carried out in refluxing chloroform and YbTf_3_ as the catalyst. After 24 h, the crude product, obtained upon the evaporation of the solvent, was purified by column chromatography on silica gel to afford the compound **5c** (apparently as a diastereomer mixture). The demetalation was carried out with concentrated H_2_SO_4_ (4 min at rt), and the subsequent neutralization was carried out with NaOH. The crude product obtained was then purified by column chromatography, affording compound **6c** with an overall yield of 23% as a single diastereomer (Figure 4; see Appendix A in the Appendix A). The relative stereochemistry of **6c** could not be determined by ^1^H-^1^H NOESY or by ^1^H-^1^H ROESY NMR. However, as in the case of the compounds **6a** and **6b’**, taking again into account the known preference for the formation of *trans*-imidazolidinones, [18,41] we assigned a *trans* relationship between the two substituents at C2 and C5 in the imidazolidinone ring.

In summary, starting with the Cu(II) complex **3**, we were able to obtain three new tetraphenylporphyrin–imidazolidinone hybrids of Type I (**6a**, **6b**, and **6c**) through Ytterbium(III) catalyzed condensation with the suitable *N*-methyl amides **4a**, **4b**, and **4c**. Two hybrids (**6a** and **6c**) were obtained as single stereoisomers. In the case of **6b**, in which two stereoisomers were isolated, we could safely assign a *trans*-relative stereochemistry to the major one. The critical step in this sequence was the demetalation, which was accompanied by the hydrolysis of the imidazolidinone ring.

The synthesis of 5-(4-formylphenyl)-10,15,20-triphenylporphyrin **2**, the porphyrin building block for the construction of bifunctional photocatalysts of Type B, began with the preparation of 5-(4-(methoxycarbonyl)phenyl)-10,15,20-triphenylporphyrin **8** through a mixed porphyrin synthesis by the reaction of one equivalent of methyl-4-formylbenzoate **7** with three equivalents of benzaldehyde and four equivalents of pyrrole, in refluxing nitrobenzene in the presence of acetic acid [44]. After chromatographic purification, the resulting monofunctionalized porphyrin, which was obtained with a remarkably high yield (29%, corresponding to 66% of the maximum statistical yield), was reduced with lithium aluminum hydride to produce the corresponding carbinol **9**, [45] whose subsequent oxidation with pyridinium chlorochromate (PCC) afforded the target aldehyde **2** in an excellent yield. Finally, metalation with Cu(II) acetate furnished the corresponding complex **10** (Figure 5) [31].

The attempted condensation between **2** and the *N*-methyl amide **4a** (YbTf_3_, CHCl_3_, sealed tube, 24 h reflux) resulted, as in the case of **1,** in the formation of a complex mixture in which the imine **11** could be detected by NMR (singlet at 8.62 ppm in the ^1^H NMR spectrum, C**H**=N; signal at 192.4 ppm in the ^13^C NMR spectrum, CH=N). When the metalated aldehyde **10** was submitted to the same reaction conditions, the metalated imidazolidinone **12** was obtained with a 46% yield (after filtration through a silica gel pad). The treatment of this compound with concentrated sulfuric acid (3 min, rt) led to a complex reaction crude from which **13** was obtained in a low yield (27%) as a non-separable mixture of diastereomers, which could not be adequately characterized (Figure 6).

In view of these difficulties, we decided to change our strategy and prepare a porphyrin–secondary amine hybrid of Type B through the reductive amination of **2** with the L-proline-derived primary amine ((*S*)-*N*-*tert*-butyloxycarbonyl(2-aminomethyl)pyrrolidine, **18**). This compound was prepared according to the route shown in Figure 7, based on transformations previously described in the chemical literature [46]. The synthesis started with the reduction of L-proline to (*S*)-(2-pyrrolidin)methanol **14** with lithium aluminum hydride in a THF solution, followed by the subsequent *N*-Boc protection of the amine. The hydroxy group in **15** was activated via tosylate **16**, displaced with sodium azide to afford **17**, and finally reduced with triphenylphosphine in aqueous THF to the desired amine **18**.

To obtain the desired bifunctional catalyst by the reductive amination of *meso*-tetraphenylporphyrin-carbaldehyde **2** with the primary amine **18**, we started with the conditions described by Liu et al. [46] for a related transformation (MeOH/THF, NaBH_4_, reflux). However, we only observed the complete reduction of the aldehyde **2** to alcohol **9**. The same result was obtained when we used sodium cyanoborohydride in THF, or when we heated a chloroform solution of **2** and **50** in the presence of NaBH_4_/YbTf_3_.

At this point, we reasoned that the problem resided in the non-formation of the imine, and we decided to take advantage of the observed formation of the imine **11**, described above in Figure 6. Thus, **2** was reacted with 2 equiv of the *N*-Boc amine **18** in CHCl_3_ in the presence of a catalytic amount (2.4 mol%) of YbTf_3_. After heating at reflux for 24 h, TLC monitoring showed that the formation of the imine intermediate was complete. After cooling to rt, 2 equiv of sodium cyanoborohydride was added in one portion and stirring was continued for 1 h. After chromatographic purification, we were pleased to find that the desired compound, **19**, could be isolated with a 67% yield. Finally, the essentially quantitative cleavage of the *N*-Boc group was achieved by treatment with an excess of trifluoroacetic acid in a chloroform solution at rt, affording the target porphyrine–pyrrolidine hybrid **20** (Figure 8). It is worth noting that the more commonly used dichloromethane was not convenient for this purpose due to the low solubility of **19**.

When **2** was substituted by the Cu(II) complex **10**, reductive amination in the same conditions afforded the Cu(II) complex **21** with a good yield (51%, Figure 9).

In order to obtain another example of an amine-functionalized porphyrin hybrid of Type B based on the formation of an amide bond, we decided to prepare the *meso*-tetraphenylporphyrin amine derivative **24**, a compound that was first described by Bryden and Boyle [47]. To begin, the functionalized porphyrin **23**, having a nitrile group in the 4-position of one of the phenyl groups, was obtained by the mixed condensation of *p*-cyanobenzaldehyde **22**, benzaldehyde, and pyrrole in propionic acid under reflux. After two successive chromatographic purifications, **23** was obtained with a 21% yield (48% of the statistical yield). Next, the reduction of the cyano group was performed using lithium aluminum hydride, affording the desired porphyrin **24** with a 71% yield. Easily available *N*-Boc-L-proline **25** [48] was transformed to a mixed anhydride with ethyl chloroformate and triethylamine in THF at −15 °C, and a THF solution of **24** (0.83 equiv) was added dropwise. After warming to rt, the amide **26** was obtained in an almost quantitative yield following chromatographic purification. The cleavage of the *N*-Boc group by trifluoroacetic acid in chloroform provided the final porphyrin–amine hybrid **27** with a 72% yield (Figure 10).

### 2.2. Study of the Catalytic Activity of Amine–Porphyrin Hybrids of Type A in the Organocatalyzed Diels–Alder Reaction

As a preliminary study of the possible use of amine–porphyrin hybrids as asymmetric bifunctional organophotocatalysts, we selected the chiral imidazolidinone-catalyzed Diels–Alder reaction between cyclopentadiene **28** and *trans*-cinnamaldehyde **29** as a prototypical, iminium-ion-mediated enantioselective process [49]. This reaction gives rise to a mixture of *endo*- (**30**) and *exo*-adducts (**31**) that, under optimized conditions (imidazolidinone **32** [50] as the chiral organocatalyst, HCl as the acid cocatalyst, 95:5 MeOH:H_2_O), are obtained in a highly enantioenriched form (Figure 11).

To begin, we wanted to ascertain if the introduction of the bulky 5,10,15,20-tetraphenylporphyrin-2-yl substituent at the 2-position of the imidazolidin-4-one ring could negatively affect its organocatalytic activity (by inhibiting the formation of the cyclic iminium intermediate due to the increase in steric hindrance, for example). To that end, and taking as a benchmark reaction the catalysis by imidazolidinone **32** and *p*-toluenesulfonic acid in 95:5 MeOH-H_2_O (Entry 1 in Table 1), we analyzed the results obtained with our Type A imidazolidinone–porphyrin hybrids **5** and **6**. The formation of the Diels–Alder adducts was monitored by TLC and ^1^H NMR. After a partial chromatographic purification of the reaction crude (to remove unreacted **29**), the *endo:exo* ratio of the Diels–Alder adducts **30**/**31** was determined by the relative areas of the aldehydic protons in the ^1^H NMR spectrum (9.60 and 9.92 ppm in CDCl_3_, respectively; see Appendix A). The adduct mixture was reduced to the corresponding alcohols by the procedure of Hayashi and co-workers (NaBH_4_, MeOH, rt, 24 h) [51], and the enantiomeric composition was determined by chiral HPLC (see Appendix A) [25].

When the metalated imidazolidinone **5a** (derived from L-alanine, see Figure 2) was used as a catalyst, we observed that it was only partially soluble in 95:5 MeOH:H_2_O, and that the isolated yield of the adducts was only 17% after 72 h (Entry 2 in Table 1). The *endo/exo* ratio and the enantiomeric purities were also very different from those observed for the reference reaction with imidazolidinone **32**. When the reaction was performed in toluene, in which the porphyrine complex **5a** was more soluble, the yield increased to 30% (entry 3). In this case, the *endo* adduct **30** was the major product; however, both adducts were obtained in essentially racemic form. With the metalated porphyrin **5b**, which was derived from L-phenylalanine (Figure 3), the reaction could be performed in aqueous MeOH with a global yield of 36% (Entry 4 in Table 1). In this case, the *exo* adduct **31** was the main product, which was obtained with a moderate ee (43%). The *endo* isomer **30** showed a similar enantiomeric purity (37% ee). However, as **5b** was obtained as a *cis*/*trans* diastereomer mixture of unknown composition, these results are difficult to rationalize. We then redirected our attention to the non-metalated porphyrins **6**. The bifunctional porphyrin **6a** was not soluble in aqueous MeOH, and only traces of product were observed after 72 h (Entry 5 in Table 1). We then evaluated the organocatalytic activity of the two *cis* and *trans* isomers of **6b** in toluene. The *cis* isomer **6b** (Entry 6) gave a poor yield (9%), with a stereoselectivity (3:1 *endo*:*exo*, essentially racemic products) very similar to that observed for **5a** in the same solvent (Entry 3). On the other hand, the *trans* isomer **6b’** (Entry 7 in Table 1) provided a higher yield (27%) with a similar diastereoselectivity (*ca.* 70:30 *endo*:*exo*). While the *endo* adduct was obtained with a very low ee (3% ee), the *exo* was formed with a nearly 50% ee. The *tert*-leucine-derived porphyrin–imidazolidinone hybrid **6c** (Figure 4) did not catalyze the reaction, apparently due to steric hindrance (Entry 8). In summary, while the introduction of the 5,10,15,20-tetraphenylporphyrin-2-yl moiety had a significant impact on both the yield and in the stereochemical outcome of the Diels–Alder cycloaddition, our imidazolidinone–porphyrin hybrids of Type A still present a significant organocatalytic activity in most cases (except for the *tert*-leucine-derived compound **6c**).

In order to evaluate the impact of photosensitization on the imidazolidinone-catalyzed Diels–Alder cycloaddition of Figure 11, we decided to perform the reaction under visible light irradiation using both 5,10,15,20-tetraphenylporphyrin (TPPH_2_) and 5,10,15,20-tetrakis(4-sufonatophenyl)porphyrin (as the sodium salt Na_4_TPPH_2_S_4_ or in zwitterionic form (H_3_O)_2_TPPH_4_S_4_) [25] as photocatalysts. Additionally, (*S*)-5-benzyl-2,2,3-trimethylimidazolidin-4-one **32**, (*S*)-2,2,3,5-tetramethylimidazolidin-4-one **33** [50], and (2*S*,5*S*)-5-benzyl-2-*tert*-butyl-3-methylimidazolidin-4-one **34** [43] were used as organocatalysts (Figure 3). The results of these dual organophotocatalyzed Diels–Alder reactions are summarized in Table 2.

When the reaction was performed in an MeOH solution with imidazolidinone **32** as the organocatalyst (i.e., under the conditions of the benchmark reaction, Figure 11 and Entry 1 of Table 1) but under white light irradiation in the presence of *meso*-tetraphenylporphyrin (5 mol%), the stereochemical outcome (dr, %ee for both adducts) of the reaction was essentially the same as in the purely thermal reaction (Entry 1 of Table 2). However, we were pleased to find that the mixture of adducts **30**/**31** was obtained in an essentially quantitative yield after chromatographic purification. It is important to note that this was not due to external heating of the solution (see Experimental Section). A drastic change was observed when toluene was used as a solvent (Entry 2); in this case, no reaction was observed, probably due to the very low solubility of the *p*-toluenesulfonate salt of **32**. When the zwitterionic form of 5,10,15,20-tetrakis(4-sulfonatophenyl)porphyrin ((H_3_O)_2_TPPH_4_S_4_) was used as a photocatalyst in MeOH solution, the results were also very similar to those obtained in Entry 1 with TPPH_2_, except that in this case, a small decrease in the enantiomeric purities were observed (Entry 3 of Table 2). The use of the basic form of the same photocatalyst in aqueous methanol (Na_4_TPPH_2_S_4,_ Entry 4 in Table 2) resulted in a much lower yield and diminished enantiomeric purities. This likely reflects the important role played by the acid co-catalyst as, in this case, the reaction was performed in the absence of *p*-toluenesulfonic acid. The replacement of the 5-benzyl group with a methyl one (imidazolidinone **33**, Entry 5 in Table 2) maintained both the yield and the *endo*:*exo* adduct ratio with respect to Entry 1; a small decrease in the enantiomeric purity took place, in accordance with the role attributed by MacMillan to the benzyl imidazolidinone substituent in determining the stereochemical outcome of the cycloaddition [4,49]. When the *tert*-butyl-substituted imidazolidinone **34** was used as the organocatalyst, no reaction was observed either in toluene (entry 6) or in methanol (entry 7), probably due to steric hindrance, which impeded the formation of the iminium ion intermediate. In summary, we found that the irradiation of the reaction mixture with visible light in the presence of a porphyrin photosensitizer did not appreciably change the stereochemical course of the Diels–Alder cycloaddition (as expected) but was accompanied by a great increase in the yield.

In light of these results, we finally performed the Diels–Alder cycloaddition between cyclopentadiene **28** and *trans*-cinnamaldehyde **29** under visible light irradiation, using the imidazolidinone–porphyrin hybrids of Type I **6b**, **6b’**, and **6c** as bifunctional organophotocatalysts in toluene, and in the presence of *p*-toluenesulfonic acid as a co-catalyst.

As revealed in Table 3, the resultant reactions, catalyzed by the imidazolidinone–porphyrin hybrids **6b** (*cis*, entry 1) and **6b’** (*trans*, entry 2), were similar to those obtained under purely thermal conditions: low yields and poor enantiocontrol (compared with Entries 6 and 7 in Table 1). However, it is important to note that in both instances a very high, unprecedented diastereoselectivity in favor of the formation of the *endo* adduct **30** (*ca.* 9:1) was observed. For the bifunctional catalyst **6c**, the cycloaddition did not take place (Entry 3 in Table 3), in line with the results obtained under purely thermal conditions (Entry 8 in Table 1). This was likely due to the steric hindrance of the imidazolidinone moiety.

These important differences in the stereochemical outcome of the Diels–Alder cycloaddition under visible light irradiation, using a dual catalytic system (Table 2) on one hand and a bifunctional catalyst (Table 3) on the other, suggest that, in the last case, a competition between thermal and photochemical reaction paths may take place. Under dual catalysis conditions, in which only an increase in the yield is observed, photosensitization, followed by internal conversion, may facilitate non-radiative energy transfer to key intermediates in the organocatalytic cycle. For the bifunctional catalysis, however, the drastic change observed in the stereoselectivity of the process (increased *endo*-selectivity and practically total loss of the enantiocontrol) suggests that the reaction may follow an alternative mechanism, involving the photoexcitation of the porphyrin moiety in the unsaturated iminium ion, followed by an intramolecular single-electron transfer via oxidative quenching to provide a cationic diradical that can trigger a radical cyclization process with the diene (Figure 12).

In conclusion, we synthesized the first examples of a novel class of amine–porphyrin hybrids. In these hybrids, as potentially active bifunctional organophotocatalysts, a chiral cyclic moiety is covalently attached to a *meso*-tetraphenylporphyrin core. In these compounds, the cyclic secondary amine moiety can be linked either to a β-pyrrolic position (Type A) or to the *p*-position of one of the *meso* phenyl groups (Type B). These two types of amine-functionalized porphyrins have been accessed from the known 2-formyl-5,10,15,20-tetraphenylporphyrin **1** (compounds **6a**, **6b**, and **6c** and the corresponding metalated Cu(II) derivatives **5a**–**c**) and from 5-(4-formylphenyl)-10,15,20-triphenylporphyrin **2** (compound **20**) through condensation or reductive amination reactions with the suitable chiral amines. An additional example of a Type B bifunctional catalyst, 5,10,15-triphenyl-20-((*S*)-4-((pyrrolidine-2-carboxamido)methyl)phenyl)porphyrin **27**, was obtained by the formation of an amide between 5-(4-(aminomethyl)phenyl)-10,15,20-triphenylporphyrin **24** and *N*-Boc-L-proline **25**. A preliminary study of the organophotocatalytic activity of the Type A amine–porphyrin hybrids, performed using the chiral imidazolidinone-promoted Diels–Alder cycloaddition between cinnamaldehyde and cyclopentadiene, demonstrated that (a) the introduction of the 5,10,15,20-tetraphenylporphyrin-2-yl substituent at the 2-position of the imidazolidin-4-one ring did not suppress the organocatalytic activity the amine, except in the case of **6c**, which has the very bulky *tert*-butyl substituent at the 5-position, (b) the irradiation of the reaction mixture with visible light in the presence of a porphyrin photosensitizer (dual organophotocatalysis) did not appreciably change the stereochemical course of the chiral imidazolidinone-catalyzed Diels–Alder cycloaddition, but was accompanied by a great increase in the yield, and (c) the bifunctional catalysis of the Diels–Alder cycloaddition under visible light irradiation resulted in a dramatic change in the stereoselectivity of the process (increased *endo*-selectivity and a practically total loss of enantiocontrol); this suggests that, in this case, the reaction may follow an alternative mechanism, involving the photoexcitation of the porphyrin moiety in the bifunctional catalyst. Further assessment of the organophotocatalytic activity of these amine-functionalized porphyrins is being performed in our laboratory.

## 3. Materials and Methods

### 3.1. General Methods

Commercially available reagents, catalysts, and solvents were used as received from the supplier. Dichloromethane for porphyrin synthesis was distilled from CaH_2_ prior to use, and THF was dried by distillation from LiAlH_4_. Deuterated solvents were supplied by Merck Life Science. For normal phase HPLC chromatography, HPLC grade solvents (hexane and isopropyl alcohol) were used directly without any purification beyond that already applied by the supplier (VWR).

Thin-layer chromatography was carried out on silica gel plates Merck 60 F_254_, and compounds were visualized by irradiation with UV light and/or and chemical developers (KMnO_4_, *p*-anisaldehyde, and phosphomolybdic acid). Chromatographic purifications were performed under pressurized air in a column with silica gel Merck 60 (particle size: 0.040–0.063 mm, Merck Life Science S.L.U., Spain) as stationary phase and solvent mixtures (hexane, ethyl acetate, dichloromethane, and methanol) as eluents.

^1^H (400 MHz) NMR spectra were recorded with a Varian Mercury 400 spectrometer (Agilent Technologies, Santa Clara, Cal, USA). Chemical shifts (δ) are provided in ppm relative to the peak of tetramethylsilane (δ = 0.00 ppm), and coupling constants (*J*) are provided in Hz. The spectra were recorded at room temperature. Data are reported as follows: s, singlet; d, doublet; t, triplet; q, quartet; m, multiplet; br, broad signal. IR spectra were obtained with a Nicolet 6700 FTIR instrument (Thermo Fisher Scientific, Waltham, MA, USA), using ATR techniques. UV–vis spectra were recorded on a double-beam Cary 500-scan spectrophotometer (Varian); cuvettes (quartz QS Suprasil, Hellma, Hellma GmbH & Co. KG, Mülheim, Germany) cm were used for measuring the absorption spectra. The porphyrin solutions in water were carefully degassed by gentle bubbling a nitrogen gas stream prior to the spectrophotometric measurement.

The chiral HPLC analyses of the Diels–Alder reaction products were performed on a Shimadzu instrument (Shimadzu Europa GmbH, Essen, Germany) containing a LC-20-AD solvent delivery unit, a DGU-20AS degasser unit, and a SPD-M20A UV/VIS Photodiode Array detector with a chiral stationary phase (250 mm × 4.6 mm Phenomenex^®^ i-cellulose-5 column; Phenomenex España, S.I., C. de Valgrande 8, Alcobendas, 28018, Madrid, Spain). All solvents were of HPLC grade and were carefully degassed prior to use. The sample was injected at time 0.

*Meso*-tetraphenylporphyrin TPPH_2_, [52] 5,10,15,20-tetrakis(4-sufonatophenyl)porphyrin tetrasodium salt Na_4_TPPH_2_S_4_, [25] amides **4a**, [41,53] **4b**, [41] and **4c**, [43] *N-*Boc-(*S*)-2-(aminomethyl)pyrrolidine **18**, [46] *N*-Boc-L-proline **25**, [48] and imidazolidinones **32**, [50] **33**, [50] and **34**, [43] were prepared according to previously described procedures. See the Appendix A for more details.

### 3.2. Synthetic Procedures and Product Characterization

#### 3.2.1. Synthesis of Porphyrin-Derived Imidazolidinones

Copper(II) meso-tetraphenylporphyrin (TPPCu). In a 250 mL round-bottomed flask, equipped with magnetic stirring and a Dimroth reflux condenser, 5,10,15,20-tetraphenylporphyrin TPPH_2_ (450 mg, 0.7 mmol) was dissolved in DCM (60 mL). Once all the solid was dissolved, MeOH (20 mL) and Cu(OAc)_2_·H_2_O (243 mg, 1.2 mmol) were added, and the reaction mixture was heated up to reflux and stirred for 2 h until all the starting material was consumed (one spot in TLC, Hexane/DCM (1/1)). The solvent was then removed by distillation under reduced pressure, and the residue was redissolved in the minimum amount of DCM and filtered through a short plug of silica gel, using DCM as an eluent. After filtration, the solvents were evaporated under reduced pressure to afford the desired compound as a purple solid (432 mg, 97% yield), which was not characterized [40].

2-Formyl-5,10,15,20-tetraphenylporphyrin (1). Dry N,N-dimethylformamide (11.8 mL, 152.04 mmol) was placed in a 500 mL round-bottomed flask under argon and cooled in an ice bath. Phosphorus oxychloride (9.3 mL, 99.12 mmol) was added slowly, forming a viscous, golden mixture containing the Vilsmeier–Haack reagent. In another 500 mL round-bottomed flask, copper(II) meso-tetraphenyl porphyrin TPPCu (1.05 g, 1.55 mmol) was stirred with dry 1,2-dichloroethane (105 mL) under argon and cooled in ice. The porphyrin solution was added to the Vilsmeier–Haack reagent, and the argon source was replaced with a drying tube before warming the mixture to room temperature. It was then heated at reflux for 7 h. The reaction was cooled overnight before adding concentrated sulfuric acid (19.2 mL, 360.8 mmol) to the vigorously stirred mixture. After stirring for 10 min, the green two-phase mixture was poured into ice-cold aq. sodium hydroxide (30.2 g in 1.05 L) in a separating funnel. Chloroform (0.59 L) was then added. The mixture was shaken until no green color remained or reappeared. The bottom layer was separated and washed twice with saturated aqueous NaHCO_3_ (2 × 0.42 L). The organic layer was then dried with anhydrous MgSO_4_ and the solvent was evaporated in vacuum. The solid was filtered through a silica pad with DCM as eluent to produce 0.863 g (87% yield) of the desired product **1** [30].

Purple solid. **^1^H NMR** (400 MHz, CDCl_3_) δ 9.41 (s, 1H, C**H**O), 9.24 (s, 1H), 8.93–8.85 (m, 4H), 8.78–8.77 (m, 2H), 8.28–8.15 (m, 8H), 7.85–7.75 (m, 12H), −2.54 (s, 2H) ppm.

**Copper (II) 2-formyl-5,10,15,20-tetraphenylporphyrinate (3).** In a 500 mL round-bottomed flask equipped with magnetic stirring and a Liebig reflux condenser, under an Ar atmosphere, dry N,N-dimethylformamide (11.1 mL, 154 mmol) was added and cooled down in an ice-water bath. Then, phosphorous oxychloride (8.5 mL, 91.8 mmol) was added slowly, showing the formation of the Vilsmeier–Haack complex as a viscous, golden mixture. In another 500 mL round-bottomed flask, equipped with magnetic stirring, copper(II) meso-tetraphenyl porphyrin **TPPCu** (1.37 g, 2.0 mmol) was added and purged with Ar. At this point, dry 1,2-dichloroethane (210 mL) was added, and the resulting suspension was stirred in an ice-water bath until no solid remained. The porphyrin solution was then added via cannula to the Vilsmeier–Haack complex, and the resulting mixture was heated up to reflux for 7 h and stirred overnight at room temperature. The reaction mixture was poured into an aqueous solution of NaOAc (105 g in 500 mL) and stirred for 10 min. After this time, the green two-phase mixture was transferred to a separatory funnel and extracted with an aqueous solution of NaOH (36 g in 1.25 L) until no green color was observed. the aqueous phase was then extracted with CHCl_3_ (3 × 200 mL), and the combined organic layers were washed with an aqueous saturated solution of NaHCO_3_ (2 × 500 mL), dried over anhydrous MgSO_4_, and evaporated under reduced pressure. The residue was redissolved in the minimum amount of DCM and filtered through a short plug of silica gel, using DCM as eluent to afford the desired product **3** (1.60 g, quantitative yield) as a purple solid [40].

Red-purple solid. **UV-vis** [toluene, λ_max_ (ε), 3.75 × 10^−6^ M]: 430 (181,400), 551 (9650), 591 (5700) nm.

**Copper(II) 2-((2R/2S,4S)-1,4-dimethyl-5-oxoimidazolidin-2-yl)-5,10,15,20-tetraphenylporphyrinate (5a).** A solution of the metalated porphyrincarbaldehyde **3** (0.50 g, 0.71 mmol), amide **4a** (0,098 g, 0.96 mmol), and Yb(SO_3_CF_3_)_3_ (5 mg, 0.008 mmol) in CHCl_3_ (7 mL) in a sealed tube was heated to reflux for 24 h. After cooling, the reaction mixture was evaporated in vacuo. It was then purified via flash column chromatography in silica gel, using DCM:MeOH (98:2) as an eluent, to afford 0.147 g (26% yield) of the desired compound **5a** as an unknown mixture of diastereomers.

Purple solid. **HRMS (ESI)** *m*/*z* calculated for C_49_H_37_CuN_6_O ( [M+H]^+^), 788.2319; found 788.2317. **UV-vis** [toluene, λ_max_ (ε), 3.75 × 10^−6^ M]: 420 (801,300), 543 (42,700), 577 (850) nm.

**2-((2R,4S)-1,4-Dimethyl-5-oxoimidazolidin-2-yl)-5,10,15,20-tetraphenylporphyrin (6a).** In a 5 mL round-bottomed flask equipped with magnetic stirring, 98% sulfuric acid (0.38 mL) was added to a vigorously stirred solution of porphyrin **5a** (25 mg, 0.316 mmol) in dry DCM (2 mL). After stirring for 4 min at rt, the green two-phase mixture was poured into ice-cold aqueous sodium hydroxide (0.576 g in 20 mL) in a separating funnel. CHCl_3_ (11.2 mL) was then added. The mixture was shaken until no green color remained or reappeared in the organic phase. The bottom layer was separated and washed twice with saturated aqueous NaHCO_3_ (2 × 8 mL). The organic layer was then dried over anhydrous MgSO_4_, and the solvent was evaporated in vacuo. The resulting solid was filtered through a silica gel pad, using DCM:MeOH (97:3) as an eluent, to produce 11 mg (48% yield) of the desired product **6a** as a single diastereomer to which a trans relative stereochemistry was tentatively assigned.

Purple solid. **^1^H NMR** (400 MHz, CDCl_3_) δ 8.85–8.75 (m, 3H), 8.69 (d, J = 4.7 Hz, 1H), 8.62 (s, 1H), 8.4 (d, J = 4.6 Hz, 1H), 8.25–8.15 (m, 8H), 8.09 (d, J = 7.4 Hz, 1H), 7.85–7.7 (m, 12H), 5.33 (s, 1H), 3.71 (q, J = 7.0 Hz, 1H), 2.78 (s, 3H), 1.24 (d, J = 7.0 Hz, 3H), −2,71 (s, 2H). **HRMS (ESI)** *m*/*z* calculated for C_49_H_39_N_6_O ( [M+H]^+^), 727.3180; found 727.3179. **UV-vis** [toluene, λ_max_ (ε), 3.75 × 10^−6^ M]: 422 (292,600), 518 (15,300), 551 (7000), 595 (5300), 651 (4200) nm.

**Copper(II) 2-(2R/2S,4S)-4-benzyl-1-methyl-5-oxoimidazolidin-2-yl)-5,10,15,20-tetraphenyl porphyrinate (5b).** Freshly prepared freshly prepared L-phenylalanine methylamide **4b** (198 mg, 1.11 mmol) was introduced in a 25 mL round-bottomed flask equipped with magnetic stirring and a Dimroth reflux condenser. A solution of copper(II) 2-formyl-5,10,15,20-tetraphenyl porphyrin **3** (960 mg, 1.36 mmol) in CHCl_3_ (12 mL) and Yb(SO_3_CF_3_)_3_ (6.0 mg, 0.010 mmol, 1 mol%) were then added sequentially. The stirred mixture was heated up to reflux for 24 h. After cooling to room temperature, the solvent was evaporated under reduced pressure and purified via flash column chromatography through Et_3_N-pretreated silica gel (2.5% *v*/*v* Net_3_). A mixture of DCM/MeOH 0.5% *v*/*v* was used an eluent. The fast-running red band, which contained unreacted starting material, was not collected. Compound **5b** (purple solid, 481 mg, 50% yield) was obtained as a diastereomeric mixture (two spots in TLC, DCM/MeOH 0.5%). It was not further characterized.

2-((2S,4S)-4-Benzyl-1-methyl-5-oxoimidazolidin-2-yl)-5,10,15,20-tetraphenylporphyrin (6b-cis) and 2-((2R,4S)-4-benzyl-1-methyl-5-oxoimidazolidin-2-yl)-5,10,15,20-tetraphenylporphyrin (6b’-trans). In a 50 mL round-bottomed flask equipped with magnetic stirring, 312 mg (0.36 mmol) of metalated porphyrin 5b was dissolved in DCM (20 mL). Concentrated H_2_SO_4_ (5 mL, 98%) was added, and the resulting green mixture was vigorously stirred for 4 min. The two-phase mixture was then poured over an ice-cold aqueous NaOH solution (9 g in 300 mL), transferred to a separatory funnel, and shaken until no green color was observed. The aqueous phase was extracted with DCM (3 × 100 mL) and the combined organic layers were washed with an aqueous saturated solution of NaHCO_3_ (2 × 100 mL) and dried over anhydrous MgSO_4_. The solvent was evaporated under reduced pressure to afford the crude product, which was purified via flash column chromatography through Et_3_N-pretreated silica gel (2.5% *v*/*v* NEt_3_), using DCM/MeOH 2% as an eluent. In this way, the pure *cis* (purple solid, 82 mg, 18% yield) and *trans* (purple solid, 116 mg, 26% yield) isomers of 6b could be isolated.

**6b-cis**. **^1^H-NMR** (CDCl_3_, 400 MHz): δ = 8.85–8.80 (m, 3H), 8.77 (d, J = 4.8 Hz, 2H), 8.65 (d, J = 4.8 Hz, 1H), 8.61 (s, 1H), 8.26 (d, J = 7.4 Hz, 1H), 8.18 (d, J = 6.4 Hz, 4H), 8.16–8.12 (m, 2H), 8.10 (d, J = 7.4 Hz, 1H), 7.81–7.69 (m, 11H), 7.68 (dd, J = 7.6 Hz, J’ = 1.3 Hz, 1H), 7.33–7.19 (m, 5H), 5.34 (s, 1H), 3.93 (dd, J = 9.1 Hz, J’ = 3.6 Hz, 1H), 3.08 (dd, J = 14.0 Hz, J’ = 3.6 Hz, 1H), 2.78 (dd, J = 14.0 Hz, J’ = 9.1 Hz, 1H), 2.74 (s, 3H), 2.16 (br, 1H), −2.73 (br, 2H) ppm. **^13^C-NMR** (CDCl_3_, 100 MHz): δ = 174.6, 142.2, 142.1, 142.0, 141.9, 141.82, 142.76, 138.2, 134.82/134.80 (2C), 134.73/134.72/134.72/134.71 (4C), 134.68/134.65/134.66 (3C), 134.6, 134.57, 133.60, 129.61/129.60 (2C), 128.9, 128.7, 128.6, 128.4, 128.04, 127.96/127.95 (2C), 126.95–126.93 (5C), 126.90/126.89/126.88 (3C), 126.8, 126.72/126.71 (2C), 120.7, 120.6, 120.5, 119.4 77.4, 77.3, 77.0, 76.6, 71.5, 37.2, 29.9 ppm. **HRMS (ESI)**: *m*/*z* calculated for C_55_H_43_N_6_O [M+H]^+^, 803.3493; found 803.3468. **UV-vis** [λ_max_ (ε), c = 2.810 × 10^−6^ M, DCM]: 421 (178,000), 451 (173,300), 517 (202,000), 552 (80,800), 593 (63,250), 650 (50,900) nm.

**6b’-trans**. **^1^H-NMR** (CDCl_3_, 400 MHz): δ = 8.84–8.79 (m, 3H), 8.77 (d, J = 4.7 Hz, 2H), 8.67 (d, J = 4.6 Hz, 1H), 8.38 (s, 1H), 8.32 (d, J = 7.5 Hz, 1H), 8.30–8.03 (m, 5H), 8.01 (d, J = 7.5 Hz, 2H), 7.86–7.70 (m, 10H), 7.67 (q, J = 7.6 Hz, 2H), 7.12–6.99 (m, 5H), 5.08 (s, 1H), 3.60 (t, J = 5.6 Hz, 1H), 3.11 (dd, J = 5.4 Hz, J’ = 1.7 Hz, 2H), 2.73 (s, 3H), 1.89 (br, 1H), −2.73 (br, 2H) ppm. **^13^C-NMR** (CDCl_3_, 100 MHz): δ = 174.1, 142.5, 142.2, 142.1, 141.94, 141.88, 139.5, 139.3, 134.9, 134.82/134.80 (2C), 134.72–134.70 (4C), 134.68/134.67 (2C), 134.63, 134.58, 134.5, 133.0, 132.5, 130.0, 129.2, 128.8, 128.7, 128.5, 128.3, 128.08, 128.05, 128.0, 127.1, 126.98–126.95 (4C), 126.96/126.95/126.94 (3C), 126.92/126.90 (2C), 126.86, 120.7, 118.5, 77.4, 77.3, 77.0, 76.7, 40.9, 29.8 ppm. **HRMS (ESI):** *m*/*z* calculated for C_55_H_43_N_6_O [M+H]^+^, 803.3493; found 803.3458. **UV-vis** [λ_max_ (ε), c = 1.338 × 10^−6^ M, DCM]: 422 (131,750), 519 (171,250), 553 (79,000), 595 (63,200), 655 (72,000) nm.

**Copper(II) 2-((2R/2S,4S)-4-(tert-butyl)-1-methyl-5-oxoimidazolidin-2-yl)-5,10,15,20-tetraphenyl porphyrinate (5c).** In a 25 mL round-bottomed flask equipped with magnetic stirring and a Dimroth reflux condenser, freshly prepared L-tert-leucine methylamide **4c** (85 mg, 0.59 mmol) was introduced. Then, a solution of copper(II) 2-formyl-5,10,15,20-tetraphenylporphyrinate **3** (318 mg, 0.45 mmol) in CHCl_3_ (4.4 mL) and Yb(SO_3_CF_3_)_3_ (3.0 mg, 0.005 mmol, 1 mol%) were added sequentially. The stirred mixture was heated up to reflux for 24 h. After cooling to room temperature, the solvent was evaporated under reduced pressure and the residue was purified using flash column chromatography through an NEt_3_-pretreated silica gel (2.5% *v*/*v* NEt_3_). A mixture of DCM/MeOH 0.5% *v*/*v* was used as an eluent. The fast-running red band, which contained unreacted starting material, was not collected. Compound **5c** (purple solid, 385 mg, quantitative yield) was obtained as a diastereomeric mixture (two spots in TLC, DCM/MeOH 0.5% *v*/*v*) and was not further characterized.

**2-((2R,4S)-4-(tert-Butyl)-1-methyl-5-oxoimidazolidin-2-yl)-5,10,15,20-tetraphenylporphyrin (6c).** In a 10 mL round-bottomed flask equipped with magnetic stirring, 385 mg of metalated porphyrin **5c** (0.46 mmol) was dissolved in 5 mL of DCM. Then, 6.5 mL (125.1 mmol) of concentrated sulfuric acid was added to the solution, which was vigorously stirred for 4 min. Then, the green two-phase mixture was poured into ice-cold aqueous NaOH (9 g in 300 mL) in a separatory funnel. The mixture was shaken until no green color remained or reappeared. The bottom layer was extracted with DCM (3 × 0.1 L) and washed twice with saturated aqueous NaHCO_3_ (2 × 0.1 L). The organic layer was then dried over anhydrous MgSO_4_, and the solvent was evaporated under reduced pressure. Finally, the obtained crude product was purified via flash column chromatography through Et_3_N-pretreated silica gel (97.5:2.5 *v*/*v*), using DCM:MeOH (97:3) as an eluent. Finally, 85 mg (23% yield) of porphyrin **6c** was obtained as a single diastereomer to which a trans relative stereochemistry was tentatively assigned.

**^1^H-NMR** (CDCl_3_, 400 MHz): δ = 8.84–8.80 (m, 3H), 8.77 (d, J = 4.8 Hz, 2H), 8.69 (s, 1H), 8.62 (d, J = 4.9 Hz, 1H), 8.34 (d, J = 6.8 Hz, 1H), 8.24–8.10 (m, 7H), 7.83–7.38 (m, 12H), 5.47 (s, 1H), 3.41 (s, 1H), 2.69 (s, 3H), 2.24 (br, 1H), 0.95 (s, 9H), −2.69 (br, 2H_1_) ppm. **HRMS (ESI):** *m*/*z* calculated for C_52_H_45_N_6_O [M+H]^+^, 769.3649; found 769.3618. **UV-vis** [λ_max_ (ε), c = 2.948 × 10^−6^ M, DCM]: 420 (284,500), 518 (14,100), 552 (5800), 593 (4300), 650 (3200) nm.

**5-(4-(Methoxycarbonyl)phenyl)-10,15,20-triphenylporphyrin (8).** In a 500 mL round-bottomed flask equipped with magnetic stirring and a Liebig reflux condenser, a mixture of glacial AcOH (200 mL) and nitrobenzene (150 mL) was heated up to reflux. Methyl 4-formylbenzoate **7** (1.44 g, 8.8 mmol) was then added in one portion. When all the solid was dissolved, benzaldehyde (2.25 mL, 21.9 mmol) and freshly distilled pyrrole (2.00 mL, 29.4 mmol) were added, and the mixture was stirred at reflux for 1 h. At this point, the solvents were distilled in vacuo and the remaining purple paste was purified via flash column chromatography, using hexane/DCM (7/3) as an eluent, to afford the desired porphyrin **8** (781 mg, 29% yield) as a purple solid [44].

**^1^H-NMR** (CDCl_3_, 400 MHz): δ = 8.87–8.78 (m, 8H), 8.42 (d, J = 8.9 Hz, 2H), 8.29 (d, J = 8.9 Hz, 2H), 8.22–8.20 (d, J = 7.4Hz, 6H), 7.79–7.72 (m, 9H), 4.10 (s, 3H), −2.77 (br, 2H) ppm.

**5-(4-(Hydroxymethyl)phenyl)-10,15,20-triphenylporphyrin (9).** In a 500 mL round-bottomed flask equipped with magnetic stirring, LiAlH_4_ (296 mg, 7.8 mmol) was added, and the system was purged with Ar. Then, dry THF (6.1 mL) was added and a solution of 5-(4-(methoxycarbonyl)phenyl)-10,15,20-triphenylporphyrin **8** (1.32 g, 1.9 mmol) in dry THF (155 mL) was subsequently added via syringe. The reaction mixture was stirred at room temperature for 1 h. At this point, the reaction was quenched with 1% aqueous HCl (211 mL), and the resulting green solution was transferred to a separatory funnel and extracted with DCM until the aqueous phase became colorless. The combined organic layers were treated with a 32% *w*/*w* aqueous solution of NH_3_, noting the changing of color from green to purple, and the aqueous phase was extracted with DCM (3 × 50 mL). The combined organic layers were dried over NaSO_4_ and concentrated under reduced pressure to affect a purple solid, which was purified via flash column chromatography, using hexane/DCM (1/1) as an eluent. The desired porphyrin **9** (1.19 g, 95% yield) was obtained as a purple solid [45].

**^1^H-NMR** (CDCl_3_, 400 MHz): δ = 8.85 (s, 8H), 8.82 (d, J = 8.0 Hz, 8H), 7.78–7.73 (m, 11H), 5.30 (s, 2H), −2.77 (br, 2H) ppm.

**4-(10,15,20-Triphenylporphyrin-5-yl)benzaldehyde (2).** In a 500 mL round-bottomed flask equipped with magnetic stirring, 5-(4-(hydroxymethyl)phenyl)-10,15,20-triphenylporphyrin **9** (395 mg, 0.60 mmol) was dissolved in freshly distilled DCM (255 mL). Pyridinium chlorochromate (395 mg, 1.8 mmol) was then added, noting a change of color from purple to dark green, and the reaction mixture was stirred for 1 h at room temperature. At this point, silica gel (60 mL) was added to the reaction crude and the solvent was evaporated under reduced pressure. The resulting dark-grey solid was purified via flash column chromatography, using DCM as eluent, to afford the desired product **2** (390 mg, quantitative yield) as a purple solid [31].

**^1^H-NMR** (CDCl_3_, 400 MHz): δ = 10.39 (s, 1H), 8.88 (AB system (part A), J = 4.8 Hz, 2H), 8.85 (s, 4H), 8.79 (AB system (part B), J = 4.8 Hz, 2H) 8.42 (d, J = 7.8 Hz, 2H), 8.29 (d, J = 7.8 Hz, 2H), 8.22 (d, J = 6.4 Hz, 6H), 7.82–7.73 (m, 9H), −2.77 (br, 2H) ppm.

**Copper(II) 5-(4-formylphenyl)-10,15,20-triphenylporphyrinate (10).** In a 50 mL round-bottomed flask equipped with magnetic stirring and a Dimroth reflux condenser, 5-(4-formylphenyl)-10,15,20-triphenylporphyrin **2** (140 mg, 0.22 mmol) was dissolved in DCM (11 mL). MeOH (4 mL) and Cu(OAc)_2_·H_2_O (79 mg, 0.39 mmol) were then added, and the resulting solution was heated up to reflux for 2 h. After this time, the solvents were evaporated under reduced pressure; the residue was redissolved in the minimum amount of DCM and filtered through a short plug of silica gel, using DCM as eluent, to afford the desired product **10** (151 mg, quantitative yield) as a purple solid [31].

**UV-vis** [DCM, λ_max_ (ε), c = 4.59 × 10^−5^ M]: 412 (479900), 537 (15800) nm.

**Copper(II) 5-(4-((2R/2S,4S)-1,4-dimethyl-5-oxoimidazolidin-2-yl)phenyl)-5,10,15-triphenyl porphyrinate (12).** L-alanine methylamide **4a** (30 mg, 0.22 mmol) was introduced to a 25 mL round-bottomed flask equipped with magnetic stirring and a Dimroth reflux condenser. Then, a solution of copper(II) 5-(4-formylphenyl)-10,15,20-triphenylporphyrinate **10** (81 mg, 0.11 mmol) in CHCl_3_ (6 mL) and Yb(SO_3_CF_3_)_3_ (0.62 mg, 1 mol%) were added sequentially. The stirred mixture was heated up to reflux for 24 h. After cooling to room temperature, the solvent was evaporated under reduced pressure and the residue was purified via flash column chromatography through Et_3_N-pretreated silica gel (2.5% *v*/*v* NEt_3_), using mixtures of DCM/MeOH (from 1% *v*/*v* to 5% *v*/*v*) as eluents. The fast-running red band, which contained unreacted starting material, was not collected. Compound **12** (40 mg, 46% yield) was obtained as a diastereomeric mixture (two spots in TLC, DCM/MeOH 0.5%) that was not further characterized.

**5-(4-((2R/2S,4S)-1,4-Dimethyl-5-oxoimidazolidin-2-yl)phenyl)-5,10,15-triphenylporphyrin (13).** In a 25 mL round-bottomed flask equipped with magnetic stirring, 40 mg (0.05 mmol) of copper(II) porphyrinate **12** was dissolved in DCM (5 mL). Concentrated H_2_SO_4_ (98%, 0.7 mL) was added, and the resulting green mixture was vigorously stirred for 3 min. The two-phase mixture was then poured over a cold aqueous NaOH solution (1 g in 340 mL), transferred to a separatory funnel, and shaken until no green color was observed in the organic layer. The aqueous phase was extracted with DCM (3 × 50 mL), and the combined organic layers were washed with an aqueous saturated solution of NaHCO_3_ (2 × 100 mL) and dried over Na_2_SO_4_. The solvent was then evaporated under reduced pressure. Finally, the obtained crude product was purified via flash column chromatography through Et_3_N-pretreated silica gel (2.5% NEt_3_ *v*/*v*), using DCM/MeOH 2% as an eluent, affording a complex diastereomeric mixture that could not be separated by column chromatography. The purple solid (10 mg, 27% yield) obtained, corresponding to compound **13**, was not characterized.

**(S)-5-(4-((((1-(tert-Butoxycarbonyl)pyrrolidin-2-yl)methyl)amino)methyl)phenyl)-10,15,20-triphenylporphyrin (19).** In a 25 mL round-bottomed flask equipped with magnetic stirring and a Dimroth reflux condenser, N-Boc-(S)-2-aminomethylpyrrolidine **18** (72 mg, 0.36 mmol) was introduced. Then, a solution of 5-(4-formylphenyl)-10,15,20-triphenylporphyrin **2** (115 mg, 0.18 mmol) in CHCl_3_ (8 mL) and Yb(SO_3_CF_3_)_3_ (2.8 mg, 0.0045 mmol, 1.2 mol%) were added sequentially. The stirred mixture was heated up to reflux for 24 h. At that point, TLC monitoring showed the complete formation of the imine intermediate. After cooling to rt, NaBH_3_CN (22.6 mg, 0.36 mmol) was added in one portion. The reaction mixture was then stirred for 1 h at rt and quenched with H_2_O (15 mL). The aqueous phase was then separated and extracted with DCM (3 × 10 mL), and the combined organic layers were dried over Na_2_SO_4_ and concentrated under reduced pressure. The resulting purple solid was purified via flash column chromatography, using a mixture of DCM/MeOH (from 1% to 5%) as an eluent, affording the desired product **19** (100 mg, 67% yield) as a purple solid.

**^1^H-NMR** (CDCl_3_, 400 MHz): δ = 8.84 (s, 8H), 8.21 (d, J = 7.7 Hz, 8H), 7.80–7.70 (m, 11H), 4.34–4.17 (m, 2H), 3.51–3.38 (m, 2H), 3.10–3.03 (m, 1H), 2.20 (m, 2H), 1.95–1.86 (m, 3H,), 1.51 (s, 11H), −2.77 (br, 2H) ppm. **HRMS (ESI):** *m*/*z* calculated for C_55_H_50_N_6_O_2_ [M+H]^+^, 827.3995.; found 827.4057.

**(S)-5,10,15-Triphenyl-20-(4-(((2-(pyrrolidin-2-yl)ethyl)amino)methyl)phenyl)porphyrin (20).** In a 25 mL round-bottomed flask equipped with magnetic stirring, (S)-5-(4-((((1-(tert-butoxycarbonyl)pyrrolidin-2-yl)methyl)amino)methyl)phenyl)-10,15,20-triphenylporphyrin **19** (100 mg, 0.12 mmol) was dissolved in CHCl_3_ (6 mL). Then, trifluoroacetic acid (6 mL) was added and the resulting green solution was stirred for 2 h at rt. At this point, the reaction was concentrated under reduced pressure, redissolved in CHCl_3_ (10 mL), and an aqueous solution of NaOH (3.45 g in 100 mL) was added, noting a change of color from green to purple. The aqueous phase was separated and extracted with DCM (10 mL portions) until colorless. The combined organic phases were dried over Na_2_SO_4_ and concentrated under reduced pressure, affording a purple solid that was purified via flash column chromatography through silica gel, using a mixture of DCM/MeOH (from 0.5 to 10%) to produce the desired porphyrin **20** (84 mg, 97% yield) as a purple solid.

**^1^H-NMR** (CDCl_3_, 400 MHz): δ = 8.82 (s, 8H), 8.20–8.16 (m, 8H), 7.76–7.69 (m, 11H), 4.41–4.27 (m, 2H), 3.49–3.34 (m, 3H), 2.32 (m, 1H), 2.17–2.07 (m, 2H), 1.82 (br, 1H), 1.31–1.25 (m, 2H), 0.88 (m, 2H) ppm. **HRMS (ESI):** *m*/*z* calculated for C_50_H_42_N_6_ [M+H]^+^, 727.3471; found 727.3549. **UV-vis** [λ_max_ (ε), 1.382 × 10^−6^ M, DCM]: 418 (533,700), 515 (49,200), 549 (22,200), 590 (5,300), 647 (5,050) nm.

Copper(II) (*S*)-5-(4-((((1-(*tert*-butoxycarbonyl)pyrrolidin-2-yl)methyl)amino)methyl)phenyl)-10,15,20-triphenylporphyrinate (**21**). This compound was obtained using the same procedure described above for the non-metalated porphyrin **19** but began with copper(II) 5-(4-formylphenyl)-10,15,20-triphenylporphyrinate 10 (73 mg, 0.10 mmol). Complex 21 (46 mg, 51% yield) was obtained as a red-colored solid that was not further characterized. The identity of 21 was confirmed through treatment with concentrated sulfuric to provide a compound identical to **20**.

**5-(4-Cyanophenyl)-10,15,20-triphenylporphyrin (23).** In a 500 mL round-bottomed flask equipped with magnetic stirring and a Liebig reflux condenser, propionic acid (350 mL) and 4-cyanobenzaldehyde **22** (1.93 g, 14.7 mmol) were added sequentially. Once all the aldehyde was dissolved, the mixture was heated up to reflux and benzaldehyde (4.5 mL, 44.1 mmol) and freshly distilled pyrrole (4.1 mL, 58.8 mmol) were added. The black reaction mixture was stirred for 1 h under reflux and was protected from light. The reaction mixture was then cooled down to rt and the solvent was removed via vacuum distillation. Next, the obtained purple paste was purified by flash column chromatography through silica gel, using DCM as an eluent. In this way, the less-polar **TPPH_2_** could be separated from another fraction containing the mixture of substituted porphyrins. After the elimination of the solvent, the crude mixture was purified again by flash column chromatography in silica gel, using DCM/Hexane (4/1) as an eluent, affording the desired product **23** (868 mg, 21% yield) as a metallic purple solid [47].

**^1^H-NMR** (CDCl_3_, 400 MHz): δ = 8.89–8.72 (m, 8H), 8.33 (d, J = 7.9 Hz, 2H), 8.25–8.18 (m, 6H), 8.06 (d, J = 8.2 Hz, 2H), 7.80–7.70 (m, 9H), −2.77 (br, 2H) ppm.

**5-(4-(Aminomethyl)phenyl)-10,15,20-triphenylporphyrin (24).** In a 500 mL round-bottomed flask equipped with magnetic stirring, LiAlH_4_ (310 mg, 8.2 mmol) was introduced, and the system was purged with Ar. Then, dry THF (6.1 mL) was added and, subsequently, a solution of 5-(4-cyanophenyl)-10,15,20-triphenylporphyrin **23** (868 mg, 1.4 mmol) in dry THF (155 mL) was added via syringe. The reaction mixture was stirred at room temperature for 1 h. At this point, the reaction was quenched with aqueous 1% HCl (211 mL), and the resulting green solution was transferred to a separatory funnel and extracted with DCM until colorless. The combined organic layers were treated with a 32% (*w*/*w*) aqueous solution of NH_3_, which brought about a change of color from green to purple. The aqueous phase then was separated and extracted with DCM (3 × 50 mL). The combined organic layers were dried over Na_2_SO_4_ and concentrated under reduced pressure to afford a purple solid. The purple solid was purified via flash column chromatography through silica gel, using a mixture of DCM/MeOH (from 2% to 5%) as an eluent. The desired porphyrin **24** (618 mg, 71% yield) was obtained as a purple solid [47].

**^1^H-NMR** (CDCl_3_, 400 MHz): δ = 8.83 (m, 8H), 8.26 (br, 2H) 8.25–8.10 (m, 8H), 7.85–7.65 (m, 11H), 4.25 (s, 2H), −2.67 (br, 2H) ppm.

**(S)-5-(4-((1-(tert-Butoxycarbonyl)pyrrolidine-2-carboxamido)methyl)phenyl)-10,15,20-triphenyl porphyrin (26).** In a 25 mL round-bottomed flask equipped with magnetic stirring, a solution of N-Boc-L-Proline **25** (251 mg, 1.2 mmol) and NEt_3_ (270 mL, 2.1 mmol) in dry THF (2.1 mL) was cooled down to −15 °C. A solution of ethyl chloroformate (110 mL, 125 mg, 1.15 mmol) in dry THF (1.6 mL) was then added, and the mixture was stirred for 30 min at −15 °C. Next, a solution of 5-(4-(aminomethyl)phenyl)-10,15,20-triphenylporphyrin **24** (618 mg, 0.96 mmol) in dry THF (5.1 mL) was added dropwise, and the reaction mixture was stirred overnight at room temperature. At this point, DCM (15 mL) was added. The organic phase was then washed with NaHCO_3_ (15 mL) and brine (15 mL), dried over Na_2_SO_4_, and concentrated under reduced pressure, affording a purple solid that was purified by flash column chromatography through silica gel, using a mixture of DCM/MeOH (from 0.5% to 2%) as an eluent. The desired porphyrin **26** (806 mg, quantitative yield) was obtained as a purple solid.

**^1^H-NMR** (CDCl_3_, 400 MHz): δ = 8.84 (s, 8H), 8.22–8.16 (m, 8H), 7.76–7.63 (m, 11H), 4.93 (dd, J = 10.2 Hz, J’ = 6.4 Hz, 1H), 4.48 (m, 1H), 3.55 (m, 2H), 2.54–2.28 (m, 1H), 2.06–1.99 (m, 3H), 1.49 (s, 9H), 0.88 (m, 2H), −2.78 (br, 2H) ppm. **HRMS (ESI):** *m*/*z* calculated for C_55_H_48_N_6_O_3_ [M+H]^+^, 841.3861; found 841.3900.

**(S)-5,10,15-Triphenyl-20-(4-((pyrrolidine-2-carboxamido)methyl)phenyl)porphyrin (27).** In a 250 mL round-bottomed flask equipped with magnetic stirring, (S)-5-(4-((1-(tert-butoxycarbonyl)pyrrolidine-2-carboxamido)methyl)phenyl)-10,15,20-triphenylporphyrin **26** (806 mg, 0.96 mmol) was dissolved in CHCl_3_ (51 mL). Then, trifluoroacetic acid (51 mL) was added, and the resulting green solution was stirred for 2 h at rt. At this point, the reaction was concentrated under reduced pressure, and the residue was redissolved in CHCl_3_ (50 mL). An aqueous solution of NaOH (7.639 g in 200 mL) was then added, noting a change of color from green to purple. The aqueous phase was then extracted with DCM until colorless, dried over Na_2_SO_4_, and concentrated under reduced pressure, affording a purple solid that was purified via flash column chromatography through silica gel, using a mixture of DCM/MeOH (from 0.5% to 5%), to afford the desired porphyrin **27** (541 mg, 72% yield) as a purple solid.

**^1^H-NMR** (CDCl_3_, 400 MHz): δ = 9.09 (br, 1H), 8.82 (s, 8H), 8.21–8.14 (m, 8H), 7.74–7.65 (m, 11H), 4.97 (dd, J = 10.2 Hz, J’ = 6.4 Hz, 1H), 4.82–4.76 (m, 1H), 3.34–3.23 (m, 2H), 2.60 (br, 1H), 2.05–2.00 (m, 3H), 1.70–1.40 (m, 2H), −2.79 (br, 2H). **HRMS (ESI):**
*m*/*z* calculated for C_50_H_40_N_6_O_2_ [M+H]^+^, 741.3264; found 741.3332. **UV-Vis** [λ_max_ (ε), c = 3.892 × 10^−6^ M, DCM]: 418 (562,400), 515 (49,200), 550 (22,300), 590 (7000), 646 (5800) nm.

#### 3.2.2. Imidazolidinone-Catalyzed Diels–Alder Reactions

General procedure for the imidazolidinone-catalyzed Diels–Alder cycloaddition of cyclopentadiene 28 and (E)-cinnamaldehyde 29 under thermal conditions. The corresponding organocatalyst (0.025 mmol, 5 mol%), p-toluenesulfonic acid monohydrate (5 mg, 0.026 mmol, 5 mol%), and 1 mL of solvent (toluene or MeOH) were placed in a 3 mL vial equipped with magnetic stirring. The resulting mixture was stirred for 10 min. At this point, (E)-cinnamaldehyde 29 (63 μL, 66 mg, 0.50 mmol) and, after 30 min, freshly distilled cyclopentadiene 28 (125 μL, 98 mg, 1.48 mmol) were added, and the reaction mixture was stirred for 72 h at room temperature. The mixture was then poured over DCM (10 mL) and washed with an aqueous saturated solution of NaHCO_3_ (2 × 30 mL) until no green color remained or reappeared. The organic layer was dried over anhydrous MgSO_4_ and concentrated under reduced pressure. The ratio of the Diels–Alder adducts 30 (*endo*) and 31 (*exo*) was determined via ^1^H-NMR analysis of the crude reaction mixture (Appendix A).

When using MeOH as solvent, the Diels–Alder adducts were obtained in their acetal form, which was not purified any further. Subsequently, the deprotection was performed. The crude reaction mixture was diluted with DCM (2 mL) and TFA (1 equivalent), and H_2_O (1 equivalent) was added. After stirring for 2 h at rt, the mixture was diluted with DCM and washed with an aqueous saturated solution of NaHCO_3_ and brine. The organic layer was dried over MgSO_4_ and concentrated under reduced pressure to obtain the aldehyde adducts **30** and **31**. When using toluene as solvent, the Diels–Alder adducts were purified via flash column chromatography on silica gel, using hexane/DCM (1/1) as an eluent.

General procedure for the imidazolidinone-catalyzed Diels–Alder cycloaddition of cyclopentadiene **28** and (*E*)-cinnamaldehyde **29** under photochemical conditions. In a 3 mL vial equipped with magnetic stirring, (E)-cinnamaldehyde 29 (63 μL, 66 mg, 0.50 mmol), freshly distilled cyclopentadiene 28 (125 μL, 98 mg, 1.48 mmol), p-toluenesulfonic acid monohydrate (5 mg, 0.026 mmol, 5 mol%), and the corresponding catalyst^a^ (0.025 mmol, 5 mol%) were dissolved in 1 mL of previously degassed solvent. The vial was capped with a septum, a high vacuum was applied for 10 min, and the vial was backfilled with Ar. The process was repeated three times in order to remove all the dissolved gasses. At this point, the vial was sealed with Parafilm^®^, placed in the photoreactor (Figure 4), and stirred under irradiation for 72 h. The whole assembly was cooled by an air stream from a fan situated above the beaker with the aid of a clamp. The mixture was then diluted with DCM (10 mL) and washed with an aqueous saturated solution of NaHCO_3_ (2 × 30 mL) until no green color remained or reappeared. The organic layer was dried over MgSO_4_ and concentrated under reduced pressure. The ratio of the Diels–Alder adducts was determined via ^1^H-NMR analysis of the crude reaction mixture (Appendix A).

^a^ Employing a bifunctional catalytic system: 0.025 mmol of catalyst (5 mol%); employing a dual catalytic system: 0.025 mmol of **Na_4_TPPH_2_S_4_** or of the zwitterionic **(H_3_O)_2_TPPH_4_S_4_** as the photocatalyst (5 mol%) and 0.025 of imidazolidinone (**32**, **33** or **34**) as the organocatalyst (5 mol%).

**(1R*,2S*,3S*,4S*)-3-Phenylbicyclo [2.2.1]hept-5-ene-2-carbaldehyde (30, endo)**. Yellowish oil. **^1^H-NMR** (CDCl_3_, 400 MHz): δ = 9.60 (d, J = 2.2 Hz, 1H), 7.36–7.12 (m, 5H), 6.42 (dd, J = 5.5 Hz, J’ = 3.4 Hz, 1H), 6.18 (dd, J = 5.5 Hz, J’ = 2.7 Hz, 1H), 3.35–3.32 (m, 1H), 3.15–3.10 (m, 1H), 3.09 (d, J = 4.7 Hz, 1H), 3.00–2.95 (m, 1H), 1.81 (d, J = 9.2 Hz, 1H), 1.62 (d, J = 9.2 Hz, 1H) ppm. **HPLC** (Phenomenex i-cellulose 5 column; hexane/IPA 0.8%; flow rate 1 mL/min; 210 nm) t_R_ = 44.3 min (2R,3R), 51.6 min (2S,3S). After conversion to the corresponding alcohol with excess of NaBH_4_ in MeOH.

**(1S*,2S*,3S*,4R*)-3-Phenylbicyclo [2.2.1]hept-5-ene-2-carbaldehyde (31, exo)**. Yellowish oil. **^1^H-NMR** (CDCl_3_, 400 MHz): δ = 9.92 (d, J = 2 Hz, 1H), 7.36–7.12 (m, 5H), 6.34 (dd, J = 5.8 Hz, J’ = 3.4 Hz, 1H), 6.08 (dd, J = 5.4 Hz, J’ = 2.8 Hz, 1H), 3.73 (t, J = 3.9 Hz, 1H), 3.25–3.20 (m, 1H), 3.09 (d, J = 4.7 Hz, 1H), 3.00–2.95 (m, 1H), 2.59 (d, J = 4.9 Hz, 1H), 1.42 (d, J = 9.2 Hz, 1H) ppm. **HPLC** (Phenomenex i-cellulose 5 column; hexane/IPA 0.8%; flow rate 1 mL/min; 210 nm) t_R_ = 37.1 min (2R,3R), 48.5 (2S,3S). After conversion to the corresponding alcohol with an excess of NaBH_4_ in MeOH (See Appendix A).

## Data Availability

No new data were created in addition to those reported here and in the Appendix A.

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
