# Peer review of "Synthesis of New Amino-Functionalized Porphyrins:Preliminary Study of Their Organophotocatalytic Activity"

_molecules, 2023, doi:10.3390/molecules28041997_

Round 1

Reviewer 1 Report

This work has described the preparation of two new types of amine-porphyrin dyad derived from 5,10,15,20-tetraphenylporphyrin (TPPH2) via condensation, reductive amination or amidation reactions from the suitable porphyrin derivatives bearing either formyl or methanamine. The prepared amino-functionalized porphyrins have been investigated as a new class of bifunctional catalysts for asymmetric organophotocatalysis. This work is very interesting in the synthesis of new porphyrin based photocatalysts and the draft is well organized. This reviewer recommend its publication after minor revision.

1.      The title, “novel” is not encouraged to be used for many journal, because of it is used too frequently before. Suggest replace it by “new” or delete it.

2.      From table 3, it could be seen that the diastereoselectivity is in favor of the formation of endo adduct 30 (ca. 9:1), however the yield is very low. In table 2 (entry 1), very high yield could be observed. How about to use the combination of organocatalyst 32 in table 3 in methanol solvent? Is it possible to get high yield and high selectivity at the same time?

Author Response

1) According to your suggestion, we have changed the title of the manuscript to "Synthesis of new amino-functionalized porphyrins and preliminary study of their organocatalytic reactivity"

2) Please note that the results in Table 3 apply only to bifunctional organophotocatalysts 6b, 6b' and 6c. The results in Table 2 refer to  dual catalysis, in which we use both an organocatalyst (imidazolidinone 32) in conjunction with a porphyrin photocatalyst. But the reaction conditions (irradiation with white light, rt, 72 h) are exactly the same. The fact that the results are different is precisely what leads us to propose a specific reaction pathway for bifunctional organophotocatalysts (Scheme 12).

Reviewer 2 Report

In the article  entitled “Synthesis of novel amino-functionalized porphyrins  and preliminary study of their organophotocatalytic activit” by Albert Moyano and co-workers, it is  reported the development and characterization of porphyrins  linked to chiral cyclic secondary amine units at  b-pyrrolic position (Type I) or at a meso phenyl groups (Type II) using conventional chemistry. The possibility of some of these porphyrins to act as asymmetric catalysts in Diels–Alder cycloaddition between cyclopentadiene and trans-cinnamaldehyde under different conditions   was evaluated.

 The studies developed are interesting and follow the interest of the group in developing new catalytic applications of porphyrin derivatives. However, is not clear why the authors refer the synthesis of compounds at meso-positions  since their catalytic efficiency was not tested? If they were not efficient this must be mentioned in order to understand why they appear here.

Below there are some more comments that the authors must take in account before a final decision.

i)               In order to avoid any misunderstand with mechanisms involving oxygen singlet of Type I and Type II it is better to distinguish both type of derivatives using other classification.  Probably letters

ii)             The authors must correct the bond between the beta pyrrolic position and the imidazolidinone unit using a straight line. The trans configuration can be shown by introducing the hydrogen atom in the imidazolidinone unit with the adequate orientation. Check all the other structures with the same problem and do the adequate corrections. Do not forget to do the correction also in the SI.

iii)           All the information in the SI must be mentioned in the main manuscript. So, if in the manuscript is mentioned the NMR of a certain compound the figure where it is must be indicated.

iv)           The authors assigned the trans relative stereochemistry to 6a according with stereochemical preferences of 2-substituted imidazolidin-4- ones, [18,33]. Looking at the NMR in fact seems just to have one diastereomer. Did the authors try to do any further studies by NMR to confirm that?

v)             The   name of most of the porphyrin derivatives must be corrected and indicated in the usual way. This means that the new unit introduced must be indicated as substituent.  Although the chem-draw is now suggesting this type of designation,  the porphyrinic unit must be considered with the highest priority in order to facilitate the visualization of  the structures from the name.

vi)           The authors refer that in the Supplementary Materials is justified the relative cis stereochemistry to the minor isomer 6b, and a trans stereochemistry to the major isomer 6b’, on the basis of their 1H-NMR spectra. However, I could not find that justification there and based on the proton NMR of both derivatives (please refer their figure number in the main manuscript) it is not easy to visualize why one is trans and the other is cis. Why the 13C NMR has so many aliphatic carbons? It is the NMR of the mixture?

vii)         Considering that the assignment of configuration trans to the other derivatives namely 6c is based on the major formation of trans-6b’ the previous  aspect must be clarified.

viii)        The accurate mass of new compounds obtained in the phenyl group must be shown in SI. Why there is no 13 C NMR? Probably in the aliphatic region the authors could increase that part in order to facilitate the visualization of signals.

ix)            In Scheme 11 the authors could aid also the structure of the R enantiomers in order to facilitate the comprehension of the readers that are not so familiar to this type of studies.

x)             Concerning table 1 and the information there, the authors must show in SI the NMR of at least one of those reactions where the endo and exo proportion was evaluated. Also, an HPLC chromatogram must also be presented in order to show the mixture of alcohols obtained upon reduction with NaBH4/MeOH in order the readers are able to visualize the major enantiomers obtained .

xi)            As mentioned above what happen with the catalytic activity of the porphyrin hybrids of Type II?

xii)          What happen with the results obtained in Table 2 when they are performed under dark Conditions? 

xiii)        Check all the NMR and add the coupling constants to signals identified as doublet etc. Do not forget if the protons are coupling with each other they must have the same coupling. Multiplets must be referred with a range of values and not by a single value.  In my opinion some of the protons in the aliphatic region should be identified since the introduction of the extra unit is not so complex.

xiv)        Sometimes in the 13C NMR the authors indicate a signal and refer two carbons or more. What this mean?  if this happen  these carbons can be  indicated with two digits after comma.

xv)          Correct also typos like porphyrine to  porphyrin or porphyrins. The name of Copper(II) complexes. can be referred  as copper(II) complex of  meso-tetraphenylporphyrin  or using the termination porphyrinate. 

xvi)          The authors must improve  the introduction with the addition of more references  involving other reactions of  beta formyl-TPP  where similar condensations were performed. It is also important  to explain why  the  selection of that group.

Author Response

  1. In order to avoid any misunderstand with mechanisms involving oxygen singlet of Type I and Type II it is better to distinguish both type of derivatives using other classification. Probably letters

We agree with the referee. We have replaced “Type I” by “Type A” and “Type II” by “Type B” throughout the manuscript.

  1. The authors must correct the bond between the beta pyrrolic position and the imidazolidinone unit using a straight line. The trans configuration can be shown by introducing the hydrogen atom in the imidazolidinone unit with the adequate orientation. Check all the other structures with the same problem and do the adequate corrections. Do not forget to do the correction also in the SI. 

We have made the changes in the drawings suggested by the referee, both in the main text and in the Supplementary Material (SM).

  • All the information in the SI must be mentioned in the main manuscript. So, if in the manuscript is mentioned the NMR of a certain compound the figure where it is must be indicated.

We have numbered all Figures in the SM so that NMR of compounds mentioned in the main text always refer to the corresponding Figure in the SM

  1. The authors assigned the trans relative stereochemistry to 6a according with stereochemical preferences of 2-substituted imidazolidin-4- ones, [18,33]. Looking at the NMR in fact seems just to have one diastereomer. Did the authors try to do any further studies by NMR to confirm that?

We performed 2D NMR experiments (ROESY, NOESY) both for 6b, 6b’ and 6c, but these experiments did not reveal any interaction between the substituents at the C2 and C5 positions. This is mentioned both in the main text and in the SM.

  1. The   name of most of the porphyrin derivatives must be corrected and indicated in the usual way. This means that the new unit introduced must be indicated as substituent.  Although the chem-draw is now suggesting this type of designation,  the porphyrinic unit must be considered with the highest priority in order to facilitate the visualization of  the structures from the name.

We have corrected the names of the porphyrin derivatives according to the guidelines for “Nomenclature of tetrapyrroles (recommendations 1986)” described in Eur. J. Biochem., 1988, 178, 277-328, so that the porphyrinic unit has the highest priority. Please note also that in the manuscript the ChemDraw structures of b-substituted porphyrins have been drawn in the tautomeric form that assigns the lowest number to the substituent.

  1. The authors refer that in the Supplementary Materials is justified the relative cis stereochemistry to the minor isomer 6b, and a trans stereochemistry to the major isomer 6b’, on the basis of their 1H-NMR spectra. However, I could not find that justification there and based on the proton NMR of both derivatives (please refer their figure number in the main manuscript) it is not easy to visualize why one is trans and the other is cis. Why the 13C NMR has so many aliphatic carbons? It is the NMR of the mixture?

Pages ESI2 to ESI4 in the SM are devoted to a detailed explanation of our assignation of the relative stereochemistry of compounds 6b and 6b’, based in 1H NMR spectra (Figure S1), on their conformational preferences (showcasing a restricted rotation for the benzyl group for the cis isomer, due to a p-stacking interaction between the Phe and the porphyrin moieties, Figure S2), and supported by MMFF calculations (Figure S3).

With regard to the 13C NMR spectra of compounds 6b and 6b’, some signals (especially those with chemical shifts lower than 30 ppm) are due to solvent impurities from the chromatographic separation. They have been removed from the description of the spectra. We apologize for this oversight.

  • Considering that the assignment of configuration trans to the other derivatives namely 6c is based on the major formation of trans-6b’ the previous  aspect must be clarified.

We have added a more detailed discussion at the relevant place (see paragraph above Scheme 4).

  • The accurate mass of new compounds obtained in the phenyl group must be shown in SI. Why there is no 13 C NMR? Probably in the aliphatic region the authors could increase that part in order to facilitate the visualization of signals. 

HRMS data for all previously unknown derivatives of meso-tetraphenylporphyrin (TPPH2) having a substituent in one phenyl ring (compounds 19, 20, 26, 27) are provided. HRMS of 26 had been inadvertently omitted. The identity of 21 was confirmed by conversion to 20 by treatment with concentrated sulfuric acid. The only exception is compound 13, that could not be obtained in pure form and was not characterized.

Due to the formation of aggregates, we were not able to obtain clean 13C NMR spectra for these compounds.

  1. In Scheme 11 the authors could add also the structure of the R enantiomers in order to facilitate the comprehension of the readers that are not so familiar to this type of studies. 

Scheme 11 has been modified to show all four possible stereoisomers of the reaction product

  1. Concerning Table 1 and the information there, the authors must show in SI the NMR of at least one of those reactions where the endo and exo proportion was evaluated. Also, an HPLC chromatogram must also be presented in order to show the mixture of alcohols obtained upon reduction with NaBH4/MeOH in order the readers are able to visualize the major enantiomers obtained.

The requested information has been added to the SM (Figures S14 and S15)

  1. As mentioned above what happen with the catalytic activity of the porphyrin hybrids of Type II?

As stated in the conclusion, the catalytic activity of the porphyrin hybrids of Type B (former Type II) is currently being evaluated in our laboratories. In this case, the benchmark reaction is the organophotocatalytic a-alkylation of aldehydes with diazoacetates (see ref. [21]). We have already found that several L-proline derivatives can be used in dual catalysis with the sulfonated porphyrin Na4TPPH2S4 as the photocatalyst, with excellent conversions and variable enantioselectivities. Compound 27 as a bifunctional organophotocatalyst has shown interesting activity in the alkylation of 3-phenylpropanal with ethyl diazoacetate (40% conversion with 5 mol% of 27, after irradiation with white light for 5 h at rt), but we have not been able for the moment to determine the enantiomeric purity of the alkylated product. Results will be reported in due course.

  • What happen with the results obtained in Table 2 when they are performed under dark Conditions?

With imidazolidinone catalyst 32, we recover the results of entry 1 in Table 1. Compound 34 does not catalyze the reaction in purely thermal conditions, probably for steric reasons.

  • Check all the NMR and add the coupling constants to signals identified as doublet etc. Do not forget if the protons are coupling with each other they must have the same coupling. Multiplets must be referred with a range of values and not by a single value.  In my opinion some of the protons in the aliphatic region should be identified since the introduction of the extra unit is not so complex.

We have checked that all multiplets indicated in 1H NMR spectra in the Experimental Section (and in the SM) have the corresponding J values, and (within 0.1 Hz) coupled protons have the same J values. All multiplets are now referred to a value interval.

  • Sometimes in the 13C NMR the authors indicate a signal and refer two carbons or more. What this mean?  If this happen  these carbons can be  indicated with two digits after comma.

This means that there are two (or more) very close signals that differ only in the second digit. This has now been indicated when appropriate.

  1. Correct also typos like porphyrine to  porphyrin or porphyrins. The name of Copper(II) complexes. can be referred  as copper(II) complex of  meso-tetraphenylporphyrin  or using the termination porphyrinate. 

Nomenclature issues have been corrected and clarified as explained under issue v) above.

  • The authors must improve  the introduction with the addition of more references  involving other reactions of  beta formyl-TPP  where similar condensations were performed. It is also important  to explain why  the  selection of that group.

We have added a paragraph after Figure 2 in which we comment the fact that 2-formyl-tetraphenylporphyrin1 and its metal complexes have been used in several instances as convenient starting materials for the synthesis of meso-tetraphenylporphyrines directly linked with an heterocyclic moiety at the b-position, either through 1,3-dipolar cycloadditions or by lanthanum(III) triflate-catalyzed condensation reactions. We have added the most relevant references (8 new references in total). Just above Figure 2 we have emphasized the synthetic versatility of the formyl group.

We are grateful to the reviewer for his/her careful reading of our manuscript.

Round 2

Reviewer 2 Report

In general, the authors revised  the article according with the suggestions. However, figure SI must be checked since it is not visible to the readers. In order to avoid misunderstandings the authors must assign in the NMR spectra presented in SI the peaks due to solvents with an * or with the letter s. 

Author Response

We have checked that all Figures in the Supplementary Materials are visible. In order to avoid visualization problems, we upload the SM Section in pdf format.

In the 13C NMR spectra of compounds 6b-cis (Figure S6) and 6b'-trans (Figure S8) we have labelled the peks due to solvents with the letter s.